# Sustained ACE2 Expression by Probiotic Improves Integrity of Intestinal Lymphatics and Retinopathy in Type 1 Diabetic Model

**DOI:** 10.3390/jcm12051771

**Published:** 2023-02-23

**Authors:** Ram Prasad, Yvonne Adu-Agyeiwaah, Jason L. Floyd, Bright Asare-Bediako, Sergio Li Calzi, Dibyendu Chakraborty, Angela Harbour, Aayush Rohella, Julia V. Busik, Qiuhong Li, Maria B. Grant

**Affiliations:** 1Department of Ophthalmology and Visual Sciences, University of Alabama at Birmingham, Birmingham, AL 35294, USA; 2Department of Physiology, Michigan State University, East Lansing, MI 48824, USA; 3Department of Ophthalmology, University of Florida, Gainesville, FL 32611, USA

**Keywords:** intestine lymphatics, gut barrier, lipids, diabetic retinopathy, type 1 diabetes

## Abstract

Intestinal lymphatic, known as lacteal, plays a critical role in maintaining intestinal homeostasis by regulating several key functions, including the absorption of dietary lipids, immune cell trafficking, and interstitial fluid balance in the gut. The absorption of dietary lipids relies on lacteal integrity, mediated by button-like and zipper-like junctions. Although the intestinal lymphatic system is well studied in many diseases, including obesity, the contribution of lacteals to the gut–retinal axis in type 1 diabetes (T1D) has not been examined. Previously, we showed that diabetes induces a reduction in intestinal angiotensin-converting enzyme 2 (ACE2), leading to gut barrier disruption. However, when ACE2 levels are maintained, a preservation of gut barrier integrity occurs, resulting in less systemic inflammation and a reduction in endothelial cell permeability, ultimately retarding the development of diabetic complications, such as diabetic retinopathy. Here, we examined the impact of T1D on intestinal lymphatics and circulating lipids and tested the impact of intervention with ACE-2-expressing probiotics on key aspects of gut and retinal function. *Akita* mice with 6 months of diabetes were orally gavaged LP-ACE2 (3x/week for 3 months), an engineered probiotic (*Lactobacillus paracasei;* LP) expressing human ACE2. After three months, immunohistochemistry (IHC) was used to evaluate intestinal lymphatics, gut epithelial, and endothelial barrier integrity. Retinal function was assessed using visual acuity, electroretinograms, and enumeration of acellular capillaries. LP-ACE2 significantly restored intestinal lacteal integrity as assessed by the increased expression of lymphatic vessel hyaluronan receptor 1 (LYVE-1) expression in LP-ACE2-treated *Akita* mice. This was accompanied by improved gut epithelial (Zonula occludens-1 (ZO-1), p120-catenin) and endothelial (plasmalemma vesicular protein -1 (PLVAP1)) barrier integrity. In Akita mice, the LP-ACE2 treatment reduced plasma levels of LDL cholesterol and increased the expression of ATP-binding cassette subfamily G member 1 (ABCG1) in retinal pigment epithelial cells (RPE), the population of cells responsible for lipid transport from the systemic circulation into the retina. LP-ACE2 also corrected blood–retinal barrier (BRB) dysfunction in the neural retina, as observed by increased ZO-1 and decreased VCAM-1 expression compared to untreated mice. LP-ACE2-treated *Akita* mice exhibit significantly decreased numbers of acellular capillaries in the retina. Our study supports the beneficial role of LP-ACE2 in the restoration of intestinal lacteal integrity, which plays a key role in gut barrier integrity and systemic lipid metabolism and decreased diabetic retinopathy severity.

## 1. Introduction

The incidence of T1D is continuously increasing globally and in the United States [1], as are the ocular complications associated with T1D, including diabetic retinopathy (DR), macular edema, cataracts, and glaucoma. In the first two decades of T1D occurrence, over 90% of individuals develop DR. A critical cell in the pathogenesis of DR is the retinal pigment epithelium (RPE). RPE cells are multifunctional cells that regulate key physiological functions, including nutrient transport, light absorption, and cytokine/chemokine secretion. Impaired RPE function can lead to visual abnormalities and contribute to the pathogenesis of DR. Under the diabetic state, RPE cells are exposed not only to hyperglycemia but also to abnormal levels of circulating lipids resulting in dyslipidemia. The role of dyslipidemia and retinal lipid metabolism in diabetes remains an area of active investigation [2].

The renin–angiotensin system (RAS) is widely studied in mammals [3,4,5]. In addition to systemic RAS, there is a local RAS in tissues. Studies on T1D have shown dysregulated RAS and a decreased abundance of ACE2 [6,7,8,9,10,11,12]. ACE2, a homolog of ACE, is a monocarboxypeptidase that converts angiotensin II (Ang II) into angiotensin 1–7 (Ang 1–7) [13,14,15,16]. Previously, we have shown reduced levels of tissue-specific ACE2, such as in the retina, intestine, heart, and bone marrow of diabetic rodents and the existence of a dysregulated RAS axis in diabetes [6,7,8,9,10,11]. In the past several years, ACE2 has garnered much attention as it serves as a receptor for the entry of the SARS-COV2 virus, and it has been considered to be a therapeutic target [17,18,19,20].

The small intestine has the highest expression of ACE2 of any tissue in the body. The small intestine consists of finger-like projections known as villi. Each villus is comprised of an epithelial cell layer, enterocytes, goblet cells, enteroendocrine cells, and the lamina propria. The lacteals or intestinal lymphatics are in the central part of lamina propria and contribute to the intestine’s immune response and to the absorption of dietary lipids [21,22,23,24,25,26,27,28]. Previously, we have shown a central role of ACE2 in the intestinal pathophysiology of T1D using rodent models and human studies [11,12]. We showed that T1D individuals with DR exhibit a loss of intestinal barrier function that is identified with multiple biomarkers of gut barrier disruption, such as peptidoglycan (PGN), lipopolysaccharide-binding protein (LBP), and fatty-acid-binding protein 2 (FABP2); this increase in gut permeability was associated with gut-derived immune cell activation and, importantly, with worsening DR severity. Hyperglycemia and hyperlipidemia promote an impaired gut barrier (epithelial/endothelial) as evidenced by decreased expression of ZO-1, p-120 catenin, and VE-cadherin and fosters the increase in gut microbial peptides into the systemic circulation [11,12]. Circulating gut microbial peptides, such as PGN, reach the retina, and cause retinal vascular permeability by targeting TLR2-mediated MyD88/ARNO/ARF6 signaling and contribute to the development of a DR in T1D mice. The genetic depletion of ACE2 intensified this damage. Though the local effect of enteral ACE2 was assessed by analyzing intestinal pathophysiology, the impact of enteral ACE2 on systemic ACE2 remains unexplored [11,12]. In the present study, we explored the role of the probiotic expressing soluble ACE2 on dyslipidemia and lacteal integrity in a T1D model while evaluating the impact on DR endpoints.

## 2. Materials and Methods

### 2.1. Chemicals and Antibodies

Primary antibody specific for lacteal marker lymphatic vessel endothelial hyaluronan receptor-1 (LYVE1; #67538), p120-catenin (#59854), and Vascular cell adhesion molecule 1 (VCAM1; #39036) from cell signaling (Danvers, MA, USA), Zonula occludens-1 (ZO1; #sc-33725) from Santa Cruz Biotechnology (Dallas, TX, USA), Plasmalemma vesicle-associated protein 1 (PLVAP1; #NB100-77668), and ATP-binding cassette subfamily G member 1 (ABCG1; #NB400-132) from Novus (Centennial, CO) were purchased. The ELISA assay kits for LDL-cholesterol (#79980) and HDL-cholesterol (#79990) were purchased from Crystal Chem (Elk Grove Village, IL, USA). Triglyceride assay (#10010303) and Free Fatty Acid assay (#STA-618) kits were purchased from Cayman chemicals (Ann Arbor, MI, USA) and Cell Biolabs, Inc. (San Diego, CA, USA), respectively.

### 2.2. Experimental Animals and LP-ACE2 Treatment

*Akita* mice (an animal model of T1D), originally purchased from the Jackson Laboratory (Strain#003548; Bar Harbor, ME), were housed in the animal facility at the University of Alabama (UAB) and bred using an in-house breeding scheme to generate the experimental mice. The experimental mouse colonies were maintained under standard housing conditions (12/12 h, dark and light cycle; the temperature of 24 ± 2 °C, and humidity of 50 ± 10%). The mice were fed a standard AIN76A diet and water ad libitum. The Institutional Animal Care and Use Committee (IACUC) approved the proposed study at UAB under animal protocol number 21196.

The heterozygous *Akita* mouse model is a monogenic model of T1D that develops hyperglycemia, hypoinsulinemia, polydipsia, and polyuria within 3 to 4 weeks of age [11]. As the diabetic phenotype is more severe in males than in females, only age-matched male mice were used in this study.

To determine the effect of enteral ACE2, a healthy gut bacterium, *Lactobacillus paracasei* (LP), was engineered to produce soluble humanized ACE2 (LP-ACE2) and given via gavage into the intestinal lumen, as described in Prasad et al. [12]. *Akita* mice were randomly divided into two groups: (i) mice received control probiotics (LP) and (ii) mice received LP-ACE2. Both groups were aged to 6 months. After 6 months of diabetic onset, the experimental mice were treated (200 μL, 1 × 10^10^ colony-forming units (CFU)/mouse; 3 times/week) with either LP or PL-ACE2. At the end of the study, the intestinal tissues, plasma, and eyes were collected and stored at −80 °C until further analysis. The wildtype littermates, obtained after the genotyping of the *Akita* mice, were used as age-matched controls [12].

### 2.3. Assessment of Retinal Functions and Acellular Capillaries Quantification

To determine the effect of 3 months of treatment (LP or LP-ACE2), retinal function was assessed by electroretinography (ERG) and visual acuity by optokinetic nystagmus (OKN). All of the cohorts underwent dark adaption, and then ERGs were performed using the LKC Bigshot ERG system [29,30]. Scotopic rod signaling was assessed with 10 increasing intensities of white light, and the responses were averaged and analyzed using the LKC EM software. Acellular capillaries were assessed to evaluate the effect of LP-ACE2 on DR phenotype, as reported by Asare-Bediako et al. [29].

### 2.4. Circulating Lipid Profile Assay

Circulating levels of LDL-cholesterol, HDL-cholesterol, Triglycerides (TGA), and free fatty acids (FFA) were measured in the plasma samples of all three cohorts, followed by the manufacturer’s protocol.

### 2.5. Immunofluorescence Staining of Intestinal Lacteal, Gut Barriers, and Retina

To determine the effect of LP-ACE2 on intestinal lymphatic, gut barrier integrity, and retinal damage in *Akita* mice, immunofluorescence staining was performed as described previously [12] using the intestinal lacteal marker LYVE1, the gut endothelial marker PLVAP1 and epithelial makers (ZO-1 and P120 catenin) specific antibodies. Retinal damage was assessed by the immunostaining of ZO-1 and VCAM1-specific antibodies. The expression of reverse cholesterol efflux transporter, ABCG1, was stained in the retina. Briefly, paraffin sections were deparaffinized, epitope retrieved, dehydrated, and then incubated with specific primary antibodies (1:200) overnight at 4 °C. After overnight incubation, the sections were washed in PBS and incubated with secondary color-conjugated antibodies at room temperature for 2 h. Images were collected using a Nikon A1R-HD confocal microscope equipped with NIS-Elements AR Software and a Zeiss Axio epifluorescence microscope. Nuclei were stained with DAPI (blue) in all immunofluorescence images. The fluorescence intensity was quantified by measuring mean gray intensity using ImageJ software (Java 8). Representative images for immunofluorescence staining were selected from a data point closest to the mean or median intensity from its representation.

### 2.6. RNA Isolation and qRT-PCR

The total RNA was extracted from retinal tissue using QIAGEN RNeasy Plus Micro Kit (#74004; Germantown, MD, USA), followed by cDNA synthesis using Bio-Rad iScript cDNA synthesis kit (#1708890). qRT-PCR for primers specific to ATP binding cassette family A protein 1 (ABCA1; # qMmuCID0021182) was carried out using an advanced universal SYBR supermix (#1725271; Bio-Rad, Hercules, CA, USA) following the manufacturer’s instructions. The reaction mixture contained 5 µL supermix, 1 µL primer, 1 µL cDNA, and 3 µL nuclease-free water. The PCR conditions were: 98 °C for 30 s, 95 °C for 15 s and 60 °C for 30 s (35 cycles), 60 °C−95 °C for 5 s (0.5 °C increment). The mRNA expression was normalized to cyclophilin A and presented as relative gene expression.

### 2.7. Statistical Analysis

The data were evaluated for outliers and adherence to a normal distribution using GraphPad Prism, version 8.1 software. Statistical significance was assessed one way ANOVA and Tukey’s multiple comparison test. The data sets were considered significantly different if the *p*-value was <0.05.

## 3. Results

### 3.1. LP-ACE2 Treatment Attenuates Diabetes-Induced Lacteal Defects and Gut Barrier Dysfunction in Akita Mice

We investigated whether intestinal lacteals are altered in the *Akita* mice using immunofluorescence staining of LYVE1, a widely accepted marker of lymphatic endothelium. As shown in Figure 1A, the length of lacteals was reduced in *Akita* mice. The quantification of the intensity data suggests that the expression of LYVE1 was decreased significantly in *Akita* mice (2.07 ± 0.19 vs. 5.74 ± 1.16; *p* < 0.005) compared with the WT cohort. The LP-ACE2 treatment significantly restores LYVE1 expression up to 89.96% (5.16 ± 0.43; *p* < 0.001) in *Akita* mice compared with the LP-treated Akita cohort (Figure 1B).

To maintain the lymphatic vessel integrity, lymphatic endothelial cells are connected to each other with specialized cell–cell junctions regulated by adherens junctional molecules. The disruption of these endothelial barriers impairs vessel function. Thus, we determine the effect of the LP-ACE2 treatment on PLVAP1, a marker of endothelial barrier integrity which increases with worsening disruption. The expression of PLVAP1 was strongly over-expressed in *Akita* mice (4.05 ± 1.03 vs. 1.05 ± 0.3; *p* < 0.0) compared with the WT mice (Figure 1C,D). The expression of PLVAP1 was significantly reduced (1.54 ± 0.3, *p* < 0.02) to normal levels in LP-ACE2-treated *Akita* mice.

Enterocytes, the uppermost epithelial cell layer of the villus, maintain cell–cell contact to form an interface between the intestinal lumen and internal milieu. Studies reported that 50% of the body’s cholesterol is absorbed by the intestine and passes through the barriers by diffusion from the intestine to enterocytes [26]. Impaired epithelial barriers increase paracellular transport and affect the absorption of dietary lipids. Next, the effect of LP-ACE2 treatment was tested on the integrity of the intestinal epithelial barrier. The expression of ZO-1 was significantly less in *Akita* mice (0.60 ± 0.15 vs. 3.22 ± 0.33; *p* < 0.001) compared with their WT littermates (Figure 1E,F). The oral administration of LP-ACE2 for 3 months substantially increased ZO-1 levels in *Akita* mice. Similar changes were observed in the p120-catenin expression (Figure 1E,G). Together, these results suggest that LP-ACE2 administration corrects diabetes-induced gut lymphatic barrier disruption in T1D mice.

### 3.2. Effect of Oral Administration of LP-ACE2 on Circulating Lipid Levels

To determine whether impaired lacteals and a dysfunctional endothelial and epithelial gut barrier influenced circulating lipid levels, we measured LDL cholesterol and HDL cholesterol levels in the plasma of the experimental cohorts and assessed whether LP-ACE2 administration impacted cholesterol levels. LDL cholesterol was significantly increased (72.37 ± 7.62 vs. 45.28 ± 3.72; *p* < 0.01) in *Akita* mice compared to WT, and the levels were decreased (*p* < 0.007) following LP-ACE2 treatment (Figure 2A). In contrast, the levels of HDL cholesterol were reduced considerably (30.91 ± 4.19 vs. 65.41 ± 5.12; *p* < 0.0006) in the *Akita* cohort compared to the WT mice, and HDL cholesterol increased (43.94 ± 4.83; *p* < 0.01) after LP-ACE2 treatment in *Akita* mice compared with the untreated *Akita* cohort (Figure 2B). Elevated levels of circulating triglycerides (TGs) were observed in the *Akita* mice, and in the LP-ACE- treated *Akita,* TGs were reduced (338.7 ± 22.03 vs. 190.2 ± 31.15 vs; *p* < 0.0028) (Figure 2C). Although the levels of free fatty acids (FFA) in the plasma of *Akita* mice were higher (1971 ± 298.7 vs. 882.2 ± 130; *p* < 0.02) compared with WT cohort (Figure 2D), after LP-ACE2 treatment in *Akita* mice, a nonsignificant reduction in the FFA levels was observed. Collectively these results suggest that LP-ACE2 corrects key dyslipidemia endpoints in *Akita* mice.

### 3.3. LP-ACE2 Restored Reverse Cholesterol Transport Protein (ABCG1/ABCA1) in the Retina of Akita Mice

We next examined if LP-ACE2 administration could modulate retinal lipid homeostasis, a process largely regulated by RPE cells. RPE cells in the posterior retina are responsible for the phagocytosis of lipid-rich photoreceptors and for the uptake of circulating lipids from the choriocapillaris [31]. To protect against excess intracellular accumulation, RPE cells recycle these lipids and export them with the help of the ABCA1/ABCG1efflux pathway [31,32,33]. Thus, we determined the effect of LP-ACE2 treatment on reverse cholesterol transport proteins. The expression of ABCG1 was strongly decreased in *Akita* mice (783.0 ± 87.94 vs. 1113.0 ± 64.24; *p* < 0.04) compared with WT mice (Figure 3A,B). The expression of ABCG1 was significantly restored (1273.0 ± 124.5, *p* < 0.007) to normal levels in LP-ACE2-treated *Akita* mice. The expression of ABCA1 mRNA in the retina of diabetic and WT mice was measured. Although the mRNA expression of ABCA1 was reduced in *Akita* mice, it was not significant (Figure 3C). However, following LP-ACE2 treatment of the *Akita* cohort ABCA1 was significantly upregulated compared with untreated *Akita* mice. These results in the retina suggest that LP-ACE2 increased circulating ACE2 levels, a point which has been previously established [12]. Thus, the increasing systemic ACE2 levels likely lead to increased retinal ACE2, which increased ABCG1/ABCA1 expression and improved retinal lipid metabolism.

### 3.4. LP-ACE2 Inhibits Diabetes-Induced Blood-Retinal Barrier (BRB) Dysfunction and Microglial Inflammation in Akita Mice

BRB participates in transporting the fatty acids required for retinal function [34,35]. To assess the possible damage of BRB, the expression of ZO-1 was measured in the retina of the experimental cohorts (Figure 4A,B). While expression of ZO-1 was decreased in diabetes, the expression was restored in the LP-ACE2 treated *Akita* mice (32.41 ± 9.00 vs. 49.80 ± 7.62; ns); however, the changes did not reach significance.

VCAM-1 is a glycoprotein expressed in retinal endothelial cells. The expression of VCAM-1 is increased in the diabetic retina by pro-inflammatory cytokines, high glucose, and TLR agonists [36,37]. As seen in Figure 4A,C, retinal endothelial cells of *Akita* mice demonstrated increased VCAM-1 expression (12.33 ± 1.18 vs. 7.19 ± 1.35; *p* < 0.03) compared to WT mice. LP-ACE2 treatment decreases VCAM-1 expression below the control level (5.89 ± 0.53; *p* < 0.02) in the retina of *Akita* mice.

### 3.5. Effect of LP-ACE2 Treatment Improves Retinal Function and Reduces the Number of Acellular Capillaries in the Akita Cohort

Next, we confirmed our previous findings and showed that after 9 months of diabetes, the *Akita* mice demonstrated a significant reduction in scotopic a wave (127.8 ± 23.76 vs. 235.5 ± 29.89; *p* < 0.03) and the scotopic b wave (275.6 ± 19.05 vs. 501.2 ± 73.48; *p* < 0.02) compared to the age-matched WT mice (Figure 5A,B). LP-ACE2 treatment for 3 months significantly restored the scotopic a wave (252.7 ± 22.46; *p* < 0.01) and the scotopic b wave (534.9 ± 31.01; *p* < 0.009) in *Akita* mice compared with the *Akita* control cohort.

LP-ACE2 treatment also significantly improved visual acuity (0.41 ± 0.007 vs. 0.33 ± 0.02; *p* < 0.02) in *Akita* mice compared with the untreated *Akita* group (Figure 5C), as observed by measuring the spatial frequency.

We also enumerated acellular capillaries, a hallmark feature of DR. As shown in Figure 5D, a significantly higher number of acellular capillaries (12.33 ± 1.18 vs. 7.19 ± 1.35; *p* < 0.03) occur in the retina of *Akita* mice compared to WT mice. The increased number of acellular capillaries was almost 1.5-fold in *Akita* mice compared to their WT littermates. LP-ACE2 significantly reduced the number of acellular capillaries (5.89 ± 0.53; *p* < 0.02) in *Akita* mice.

## 4. Discussion

The salient features of our study include the demonstration that T1D promotes lacteal permeability and gut barrier defects, impairing intestinal lipid metabolism. LP-ACE2 treatment of Akita mice for 3 months corrected gut barrier defects, including improving lacteal morphology to maintain intestine lipid metabolism. LP-ACE2 significantly reduced circulating levels of LDL cholesterol, triglycerides, and FFAs, and increased HDL cholesterol. LP-ACE2 treatment repaired BRB dysfunction and reduced diabetes-induced retinal inflammation, as observed by increased ZO-1 expression and reduced VCAM-1 expression and the reduced generation of acellular capillaries. Improved visual function, as observed by measuring scotopic a and b waves and visual acuity, further demonstrated the beneficial effect of LP-ACE2. Our data support the fact that LP-ACE2, by increasing systemic ACE2 levels, improved retinal lipid metabolism and LP-ACE2 was able to restore the cholesterol efflux pathway in RPE cells by increasing the expression of ABCA1 mRNA and protein.

The small intestine is a complex organ that contributes to dietary lipid metabolism. The lacteals absorb dietary fat and regulate the intestinal immune systems. However, little is known about the function of lacteals in healthy contexts or in diabetes. Previously, we have shown that hyperglycemia promotes gut barrier permeability resulting in the release of microbial peptides into the systemic circulation and contributing to endothelial dysfunction, including in the retina [11,12]. We showed in T1D diabetic individuals that gut permeability correlates with disease severity, as the highest levels of PGN, FABP-2 and LBP were seen in the most advanced retinopathy. In our murine studies, diabetes-induced retinal endpoints were further worsened in the ACE2^-/y^ diabetic mice and maintenance of enteral ACE2 either by engineered probiotics similar to what was used in this study or by genetic overexpression of ACE2 in the gut epithelium of murine models, protected the gut barrier from the adverse consequences of diabetes [12]. We also observed that the use of LP-ACE2 improved glucose homeostasis, as measured by reduced random blood glucose and glycated A1C [12]. In contrast, although we noticed that the genetic approach of selectively expressing ACE2 in the gut epithelium of *Akita* mice corrected gut barrier defects and improved retinal function in *Akita* mice, it failed to restore normal glucose homeostasis. Thus, the ocular benefit we observed was due to ACE2 and not solely to improving glucose levels that will also prevent the progression of DR. This study brings into consideration the significance of ACE2 on lacteals function, in not only the restoration of a key aspect of the gut barrier but also by improving lipid homeostasis in the systemic circulation and in the retina.

Accumulating evidence clearly demonstrated that impaired lacteals, an essential part of the small intestine, promote diet-induced obesity and contribute to hyperlipidemia. Ablation of intestinal lacteals causes disruption of blood vessels and villi architecture, which leads to the invasion of intestinal pathogens into the circulatory system and can trigger systemic inflammation. The importance of gut lymphatics in lipid metabolism is now emerging. Numerous studies have advanced our knowledge that regulating lipid uptake by gut lymphatics and correcting lacteal defects could reverse diet-induced obesity in rodent models [38,39]. However, the role of ACE2 in protecting gut lymphatics has not been explored until the present study. The LP-ACE2 administration is known to increase circulating levels of ACE2 [30,40]; thus, we postulate that increased ACE2 in the circulation will provide more ACE2 delivery to the retina via the choriocapillaris which serves to improve the diabetes-induced reduction in the expression of lipid transporters in the RPE.

The mechanisms responsible for our observations are incompletely understood [41]. Photoreceptors in the retina demand high energy levels for their function and rely on retinal lipids for substrates for the photoreceptor’s mitochondria [42]. Though lipids are crucial for retinal function, an abnormal abundance of lipids within the retina or its surrounding environment can result in retinal dysfunction, RPE cell death, and retinal degeneration [43]. Reduced endothelial nitric oxide (NO) bioavailability and increased inducible NO drive inflammatory pathways; thus, the abnormally high retinal lipid levels promote endothelial dysfunction and likely contribute to the pathogenesis of diabetic vascular complications.

Hyperlipidemia also contributes to the pathogenesis of DR and age-related macular degeneration by endothelial dysfunction and breakdown of the BRB [44]. Individuals with higher LDL are more likely to develop retinal hard exudates than individuals with normal lipid profiles [45,46,47]. Clinical and epidemiological studies demonstrated a positive correlation between circulating levels of LDL and DR pathology [2,45,48]. Therapies lowering dietary lipids also reduce retinal hard exudates [2].

Cao et al. demonstrated a positive correlation between skeletal muscle lipid metabolism and ACE2 [49]. ACE2 deficient mice showed increased lipid accumulation in skeletal muscle, while restored endogenous ACE2 in db/db mice (an animal model of type 2 diabetes) improved lipid metabolism through the IKKβ/NF-κB/IRS-1 signaling [49]. Though the beneficial effect of ACE2 has been reported in lipid metabolism in adipose tissue, liver, and skeletal muscle, no direct evidence of ACE2 in retinal lipid metabolism has been previously reported. We reported here that systemic ACE2 improves retinal lipid metabolism. LP-ACE2, by increasing the expression of ABCG1/ABCA1, activates the cholesterol efflux pathway in RPE cells. Improved visual function and visual acuity, and reductions in acellular capillaries further demonstrated the beneficial effect of LP-ACE2.

Our study has certain limitations. The current study was limited to male mice. Our study looked at mRNA expressions for the lipid transporter; however, we did not directly measure lipid synthesis and turnover. These would represent the focus of future studies. We have previously measured systemic and ocular ACE2 levels but did not in the current study to confirm the proposed increase in both systemic and retinal ACE2 levels.

## 5. Conclusions

In summary, our studies examined intestinal lymphatics in a model of T1D and the role of ACE2 in systemic and retinal lipid homeostasis in this model. Our results demonstrate that oral administration of engineered probiotics expressing ACE2 (LP-ACE2) beneficially influences intestinal lymphatics and systemic lipid homeostasis, improving retinal function in T1D. Enteral gavage of LP-ACE2, by increasing soluble ACE2 levels in the systemic circulation, allows soluble ACE2 to be present in the choriocapillaris. When in the choriocapillaris, it is readily available to the RPE cells to improve their function and normalize lipid handling in the retina, ultimately retarding the progression of DR. Our study lends further support to the notion of using probiotics as a drug delivery platform for the management of DR in the future.

## Figures and Tables

**Figure 1 jcm-12-01771-f001:**
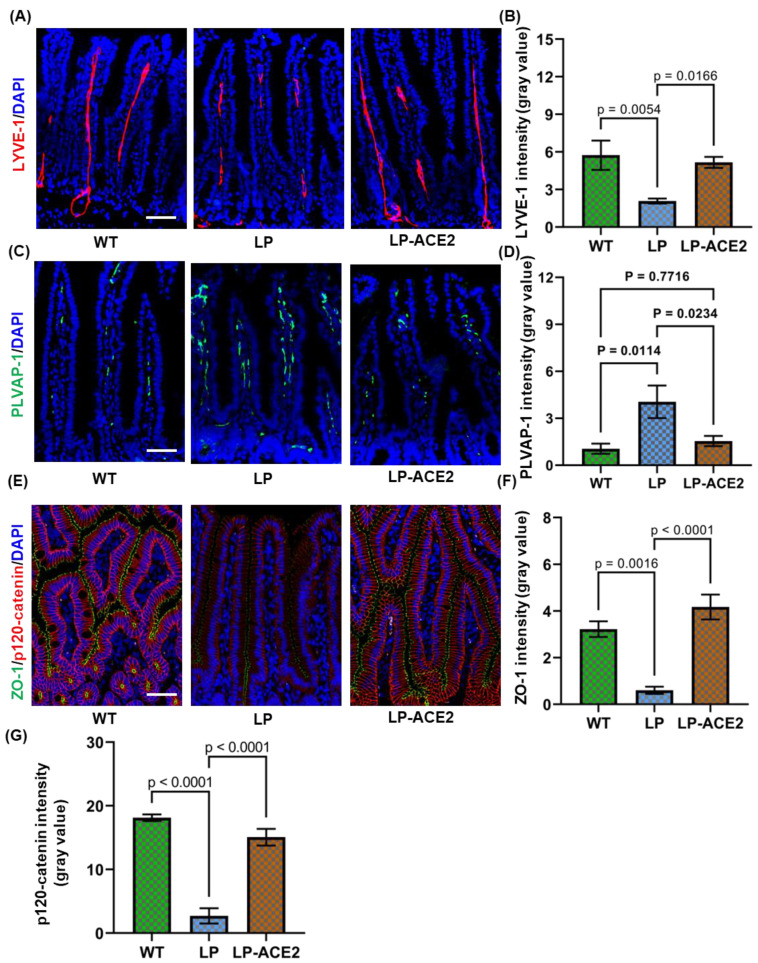
Oral administration of LP-ACE2 corrects hyperglycemia-induced lacteal disruption and improves gut barriers in *Akita* mice. Representative staining for LYVE-1 (red) in the lymphatic capillaries (lacteals) and quantification (**A**,**B**). The endothelial expression of PLVAP-1 (green) and its quantification (**C**,**D**). The intestinal barrier defects are detected by ZO-1 (green) and p-120 catenin (red) in all three cohorts (**E**). The quantification of ZO-1 and p120-catenin (**F**,**G**). The quantification of immunofluorescence staining is represented as mean intensity of gray values ± S.E.M of gray values. Nuclei were stained with DAPI (Blue). Data are analyzed by one-way ANOVA adjusted to account for multiple comparisons using Tukey’s test. Scale bar = 50 μm (*n* = 6).

**Figure 2 jcm-12-01771-f002:**
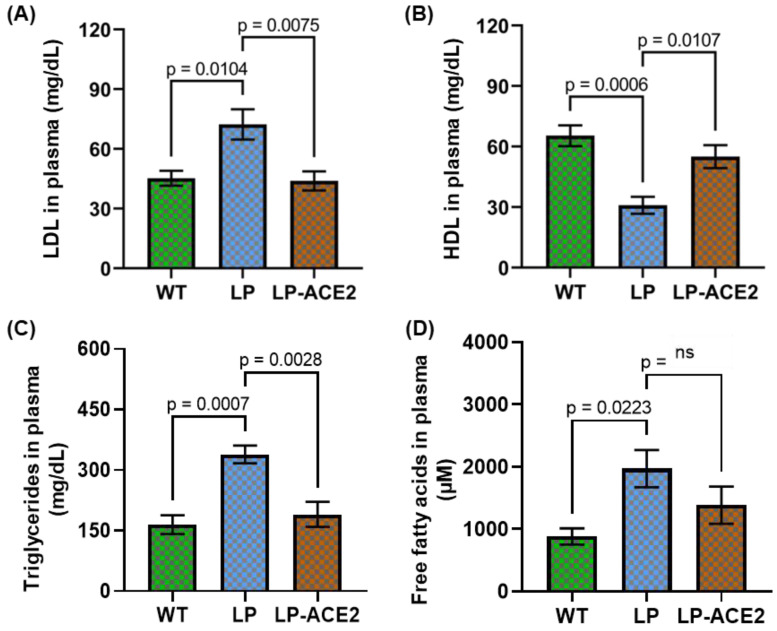
Effect of LP-ACE2 on circulating lipid profile in *Akita* and WT cohorts. The levels of LDL-cholesterol (**A**) and HDL-cholesterol (**B**), Triglycerides (**C**), and free fatty acids (**D**) were measured in the plasma samples. Data are calculated as mg/dL and presented as mean ± S.E.M. Statistical significance is analyzed by one-way ANOVA adjusted to account for multiple comparisons using Tukey’s test (*n* = 6).

**Figure 3 jcm-12-01771-f003:**
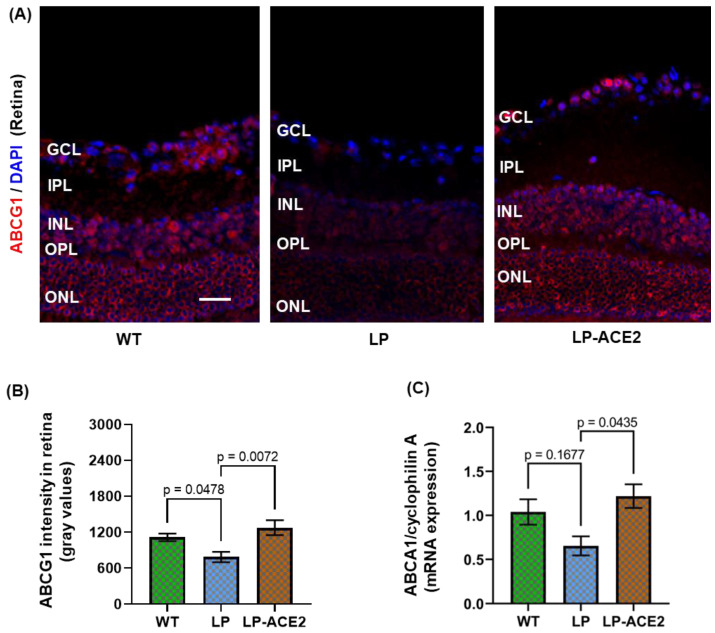
LP-ACE2 restores the reverse cholesterol transport pathway in *Akita* mice. Representative images of ABCG1 (red) expression in the retina of WT and *Akita* mice (**A**). The expression level of ABCG1 fluorescence was quantified and presented as mean intensity of gray values ± S.E.M of gray values (**B**). The mRNA expression of ABCA1 is quantified by qRT-PCR and normalized with housekeeping gene cyclophilin (**C**). Data are analyzed by one-way ANOVA adjusted to account for multiple comparisons using Tukey’s test. Scale bar = 25 μm (*n* = 6).

**Figure 4 jcm-12-01771-f004:**
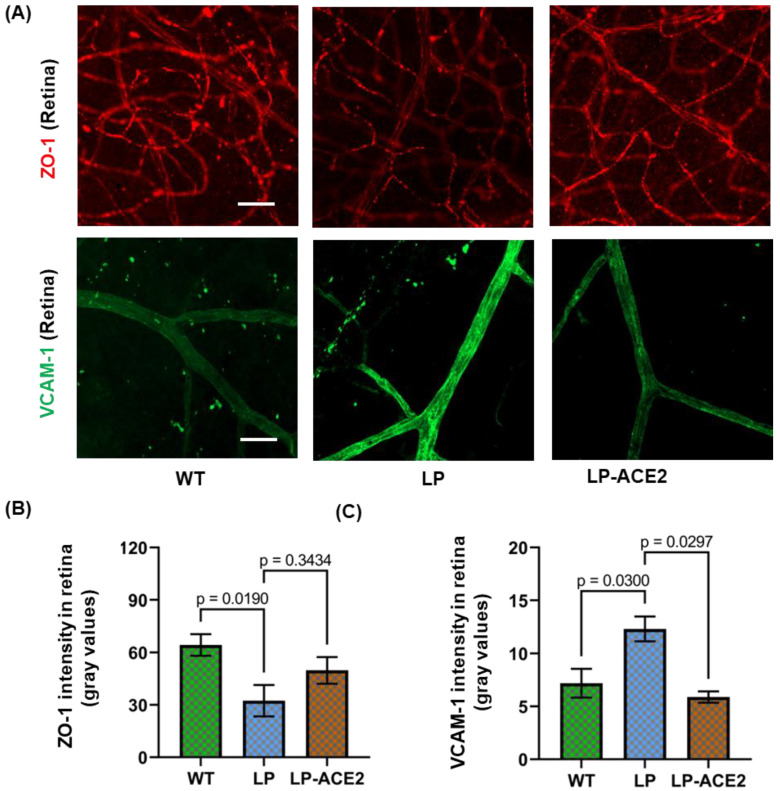
LP-ACE2 attenuates diabetes-induced disruption of blood-retinal barriers and retinal inflammation in *Akita* mice. Representative images of BRB marker ZO-1 (red) expression and retinal inflammation marker, VCAM-1 (green) expression in the retinal flatmounts (**A**). The expression levels of ZO-1 and VCAM-1 fluorescence are quantified and presented as mean intensity of gray values ± S.E.M of gray values (**B**,**C**). Data are analyzed by one-way ANOVA adjusted to account for multiple comparisons using Tukey’s test. Scale bar = 50 μm (*n* = 6).

**Figure 5 jcm-12-01771-f005:**
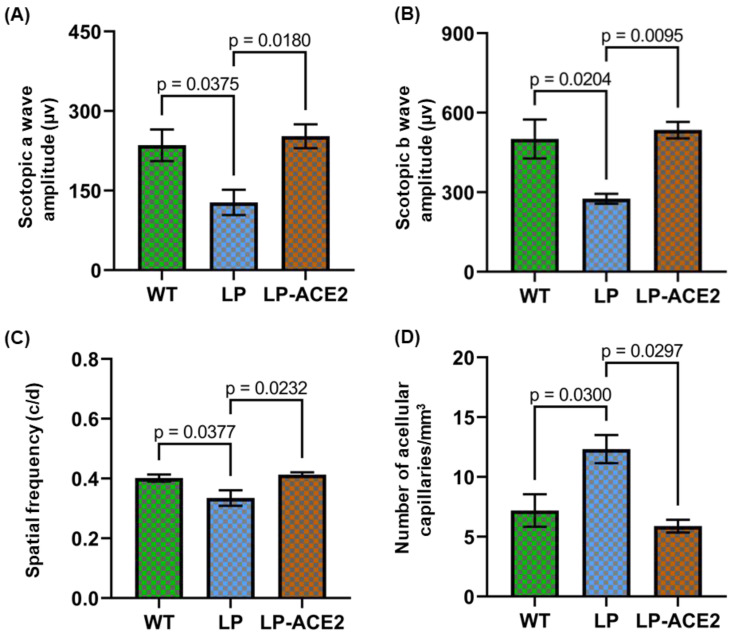
LP-ACE2 treatment corrects visual dysfunction and decreases acellular. capillary in *Akita* mice. After 3 months of LP-ACE2 treatment, the visual function in *Akita* mice was assessed by ERG and OKN. The amplitude of scotopic a-wave (**A**) and scotopic b-waves (**B**). Assessment of visual acuity using OKN demonstrates an increase in spatial frequency (**C**). In all three cohorts, the enumeration of acellular capillaries s summarized and presented as mean number of acellular capillaries/mm^3^ ± S.E.M (**D**). Data are analyzed by one-way ANOVA adjusted to account for multiple comparisons using Tukey’s test (*n* = 6).

## Data Availability

The original data presented in the study are available on request from the corresponding author.

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
