# Peer review of "Sustained ACE2 Expression by Probiotic Improves Integrity of Intestinal Lymphatics and Retinopathy in Type 1 Diabetic Model"

_jcm, 2023, doi:10.3390/jcm12051771_

Round 1

Reviewer 1 Report

1.       The abstract should be started from the general background, not with the author's own findings. Also, the authors need to compact the abstract by removing experimental details such as statistical significance. Please rewrite and update.

2.       Font style and size should be the same though out the manuscript.

3.       The conclusion is written poorly. Please rewrite and include future prospects.

4.       There are many limitations associated with this study. Please include a section entitled limitations to the present study.

5.       Which animals are used in the study, male or female? Did the authors consider any sex biases in this study? Please discuss and provide the necessary details. 

Reviewer 2 Report

Congratulations on a well done study. 

In introduction or discussion, please discuss how ace2 is relavent to diseases apart from DR. 

In introduction or discussion, please discuss how probiotics or related therapy can be used a part from the one used by your group. 

Did scotopic amplitude change before and after therapy. I  could not gather well

How is rat gut different from human, is that relevant? 

I understand that hyperglycemia was persisting when this treatment was experimented, what should happen when sugars are normalized. 
